# Analysis of Grip Amplitude on Velocity in Paralympic Powerlifting

**DOI:** 10.3390/jfmk6040086

**Published:** 2021-10-25

**Authors:** Marcelo Danilllo Matos dos Santos, Felipe J. Aidar, Andres Armas Alejo, Dihogo Gama de Matos, Raphael Fabricio de Souza, Paulo Francisco de Almeida-Neto, Breno Guilherme de Araújo Tinoco Cabral, Pantelis Theo Nikolaidis, Beat Knechtle, Filipe Manuel Clemente, Eugenia Murawska-Ciałowicz, Georgian Badicu

**Affiliations:** 1Paralympic Reference Center, Federal University of Minas Gerais (UFMG), Belo Horizonte 31270-901, Brazil; marceloed25@hotmail.com; 2Sports Training Center (CTE), Federal University of Minas Gerais (UFMG), Belo Horizonte 31270-901, Brazil; 3Department of Physical Education, Federal University of Sergipe (UFS), São Cristovão 49100-000, Brazil; fjaidar@gmail.com (F.J.A.); raphaelctba20@hotmail.com (R.F.d.S.); 4Group of Studies and Research of Performance, Sport, Health and Paralympic Sports-GEPEPS, The Federal University of Sergipe-UFS, São Cristovão 49100-000, Brazil; andymaslejo@gmail.com; 5Graduate Program of Physiological Science, Federal University of Sergipe (UFS), São Cristovão 49100-000, Brazil; 6Graduate Program of Physical Education, Federal University of Sergipe (UFS), São Cristovão 49100-000, Brazil; 7Cardiorespiratory & Physiology of Exercise Laboratory, University of Manitoba, Winnipeg, MB R3T 2N2, Canada; dihogogmc@hotmail.com; 8Department of Physical Education, Federal University of Rio Grande do Norte (UFRN), Natal 59078-970, Brazil; paulo220911@hotmail.com (P.F.d.A.-N.); brenotcabral@gmail.com (B.G.d.A.T.C.); 9School of Health and Caring Sciences, University of West Attica, 12243 Egaleo, Greece; pademil@hotmail.com; 10Exercise Physiology Laboratory, 12243 Nikaia, Greece; 11Institute of Primary Care, University of Zurich, 8091 Zurich, Switzerland; beat.knechtle@hispeed.ch; 12Medbase St. Gallen Am Vadianplatz, 9001 St. Gallen, Switzerland; 13Escola Superior Desporto e Lazer, Instituto Politécnico de Viana do Castelo, Rua Escola Industrial e Comercial de Nun’Álvares, 4900-347 Viana do Castelo, Portugal; filipe.clemente5@gmail.com; 14Instituto de Telecomunicações, Delegação da Covilhã, 1049-001 Lisboa, Portugal; 15Department of Physiology and Biochemistry, University School of Physical Education, 51-612 Wrocław, Poland; eugenia.murawska-cialowicz@awf.wroc.pl; 16Department of Physical Education and Special Motricity, University Transilvania of Brasov, 500068 Brasov, Romania

**Keywords:** Paralympic Powerlifting, grip width, relative load and velocity

## Abstract

(1) Background: Paralympic Powerlifting (PP) is a Paralympic modality that is predominantly about developing maximal force, as there are athletes who lift three times their body weight. Our objective was to evaluate the averages of the velocity for 30% and 50% of 1 Maximum Repetition (1 RM) on different amplitudes of the footprint in PP athletes; (2) Methods: The intervention happened over two weeks, with the first being devoted to the familiarization and testing of 1 RM, while in the second week, through the use of a linear Encoder, tests of velocity average (VA), velocity average propulsive (VAP), and velocity peak (VP) were carried out with loads of 30% and 50% of a maximum repetition 1 RM for 1× of the biacromial distance (BAD) 1.3 × BAD, 1.5 × BAD; (3) Results: There was a significant difference in the average velocity of 1 × BAD (1.16 ± 0.14 m/s, 1.07–1.26 IC; η2p 0.20) when compared to 1.3 × BAD (1.00 ± 0.17 m/s, 0.90–1.09 IC; η2p 0.20) over 30% of 1 RM. For the other velocity variables for 30% and 50% of 1 RM with different grip amplitudes, there were no significant differences; (4) Conclusions: In PP, the 1 × BAD footprint contributes significantly to VA at 30% of 1 RM when compared to the 1.3 × BAD and 1.5 × BAD footprints. For loading at 50% of 1 RM the VA, VAP and VP decreased when compared to 30% of 1 RM, to the extent that the VAP and VP generated with the 1.3 × BAD and 1.5 × BAD footprints were higher than those with 1 × BAD, other than for VA 50% of 1 RM, where the 1 × BAD footprint was superior to the others.

## 1. Introduction

Paralympic Powerlifting (PP) is a Paralympic modality that is predominantly about developing maximal force, with some athletes being able to lift three times their body weight [1]. The adapted supine press is the only discipline in this modality. There are some differences in this competition when compared to Conventional Powerlifting, where disabled athletes perform the movement with their legs on the bench and secured with strapping bands (optional).

As a strength modality, training in PP involves several variables, such as force, power, movement time, and bar travel velocity, among others [2]. Thus, with the adapted bench press constituting the main exercise, it is important to understand the variables involved in the execution of the movement. In this context, the load—velocity relationship has been widely used to evaluate upper body performance in the bench press [3,4]. After evaluating the force–velocity relationship for the purposes of estimating one repetition maximum (1 RM) on the basis of barbell velocity in PP athletes, Loturco et al. [5] identified a major and reliable relationship between load lifted and velocity in PP, especially at loads greater than 70% of an athlete’s 1 RM.

On the other hand, performance analysis in supine has been performed with both maximal and submaximal loads [6]. The expression of maximum force at resistances less than 1 RM is referred to as relative dynamic maximum force (rDMF) [7]. Previous studies have used relative loads to assess kinetics and kinematics in supine exercise [8]. In the study by Costa et al. [9], the mean velocity, mean propulsive velocity, and mean peak velocity were greater at 50% 1 RM when compared to 60%, 70%, and 80% 1 RM. Furthermore, relative loads of 25 to 45% of 1 RM have been good parameter settings for the assessment of training velocity and movement patterns in supine [10]. However, it is important to understand that the VA is characterized as starting from the beginning of the concentric action and continuing until the final moment of arm extension, with this being a common action to be evaluated in competition. The velocity average (VA) tends to be an important variable for measuring the velocity of displacement of the bar in the adapted supine with loads from 30 to 40% of 1 RM in PP athletes.

On the other hand, it has been observed that changes in hand spacing on the bar cause changes in the biomechanical patterns of movement and performance [11,12]. The classic study by Madsen and McLaughlin [13] identified less displacement of the bar in experienced powerlifting athletes who used wider grip widths than in non-experienced powerlifters, and therefore less mechanical work and higher performance. According to Wagner et al. [14], widths of 165 to 200% of the biacromial distance (BAD) correspond to increased force in supine when compared to 95%, 130%, 235% and 270% BAD, while 130% BAD produces more force than 95% BAD.

Most studies have compared the different BADs with respect to 1 RM strength, while little has been reported regarding the forces applied to relative loads, or to the bar travel velocity in supine [11,14]. According to Gonzalez-Badillo and Sánchez-Medina [7], the rDMF is special because it is found in most training, as well as during competitive sport gestures. Therefore, it is believed that it is necessary to study the forces applied to relative loads and the displacement velocity of the bar in PP [15,16,17,18]. Thus, the objective of this study was to evaluate the mean velocity averages for 30% and 50% of 1 RM for different handgrip amplitudes in PP athletes. It was hypothesized that the 1.5 × biacromial distance grip contributes to higher velocity, because it is the closest grip amplitude to that used by common athletes.

## 2. Materials and Methods

In the first week, the subjects were submitted to a familiarization session, 1 RM test, and definition of the biacromial distance as the base for the different amplitudes. The second week was destined to the execution of the velocity average (VA); velocity average propulsive (VAP); and velocity peak (VP) tests on 30% and 50% loads of 1 RM maximum repetition for the different grasp amplitudes. Figure 1 exemplifies the experimental design of the study.

### 2.1. Sample

The sample included 12 athletes (males) of the PP linked to a project of the Physical Education Department of the Federal University of Sergipe-Sergipe-Brazil. All participants were nationally ranked competitors eligible to compete in the sport [1], and ranked among the top ten in their respective categories. As inclusion criteria, the athletes had to have at least one year of competitive experience in the sport and to have participated in at least one competition at the national level within a period of 1 year. With respect to disabilities: four athletes presented malformation in lower limbs (arthrogryposis); two with sequelae due to poliomyelitis; and four with spinal cord injury due to accidents with injury below the eighth thoracic vertebra and two with cerebral palsy. The characterization of the sample is presented in Table 1.

The athletes participated in the study voluntarily and signed an informed consent form, according to resolution 466/2012 of the National Research Ethics Committee-CONEP, of the National Health Council, in agreement with the ethical principles expressed in the Helsinki Declaration (1964, reformulated in 1975, 1983, 1989, 1996, 2000, 2008 and 2013), of the World Medical Association. This study was approved by the Research Ethics Committee of the Federal University of Sergipe, CAAE: 2.637.882 (date of approval: 7 May 2018).

### 2.2. Procedures

In the first week, the subjects participated in a familiarization session and 1 RM test. In the second week, the collections were performed between 9:00 a.m. and 12:00 p.m. at an ambient temperature of 25 °C, depending on the subjects’ availability. During the intervention, the athletes performed a previous warm-up for the upper limbs, using three exercises (shoulder abduction with dumbbells, shoulder development on the machine, shoulder rotation with dumbbells) with three sets of 10 to 20 RM in approximately 10 min [20]. A specific warm-up was performed on the bench press itself, with 30% of the load for 1 RM, where 10 slow repetitions (3.0 × 1.0 s, eccentric x concentric) and 10 fast repetitions (1.0 × 1.0 s, eccentric x concentric) were performed to then begin the procedure.

During the test, the athletes received verbal encouragement to give maximum effort. After the end of this first stage, during the second week, the athletes performed randomly, on the basis of a draw, the VA, VAP, and VP tests, with grip widths also defined on the basis of a draw; thus, the first test was performed with 1x biacromial distance (BAD). After 48 h of rest, the test were performed with 1.3 × BAD, and following another 48 h of rest the tests were performed with 1.5 × BAD in relation to VA, VAP, and VP over loads of 30% and 50% of 1 RM. The collections took place on Monday, Wednesday and Friday, and all tests were performed with three repetitions at an interval of three to five minutes between repetitions, and then the best of the three repetitions was selected [2].

### 2.3. Instruments

The weighing of the athletes was performed on a Michetti Digital electronic platform-type Scale (Michetti, São Paulo, SP, Brazil) to facilitate their weighing while seated, with a maximum supported weight capacity of 3000 kg and dimensions of 1.50 × 1.50 m. To perform the supine exercise, an official straight bench was used (Eleiko Sport AB, Halmstad, Sweden), approved by the International Paralympic Committee [1], with 210 cm total length. The bar used was the Eleiko brand 220 cm (Eleiko Sport AB, Halmstad, Sweden) weighing 20 kg [1].

### 2.4. Load Determination 

The 1 RM test was performed, with each subject starting the trials with a weight that they believed could be lifted only once using maximum effort. Weight increments were added until the maximum load that could be lifted once was reached. If the exerciser could not perform a single repetition, 2.4 to 2.5% of the load used in the test was subtracted [21,22]. The subjects rested for 3–5 min between attempts. The test to determine 1 RM was performed one week before, at least 48 h before the evaluation process.

### 2.5. Dynamic Force Variables

Velocity average (VA), velocity average propulsive (VAP) and velocity peak (VP) were set by a dynamic measurement system, Linear Encoder (Model PFMA 3010e Muscle Lab System; Ergotest, Langesund, Norway). The encoder had its guide attached to the end of the adapted supine bar for all sets related to the loads of 30% and 50% of 1 RM [23,24]. The linear transducer automatically measured the parameters of the vertical displacement velocity of the bar at a frequency of 100 Hz for all repetitions of a set. The captured data were transferred to a personal computer and analyzed using software (Muscle Lab^®^; Ergotest, Langesund, Norway).

Average mean of the bar displacement velocity was defined from the beginning of the concentric phase to the highest moment the bar is located [25]. The mean propulsive bar travel velocity was from the beginning of the concentric phase to the moment that the bar acceleration became lower than gravity acceleration (>−9.81 m-s^−^²) [3] and the peak bar travel velocity was identified by the maximum instantaneous velocity in front of the shifted load [22,24,26].

### 2.6. Determination of Biacromial Distance (BAD)

The widths of the footprint were determined using as base the distance of the acromial processes of each athlete, and measured through a standard anthropometric device Paquimeter model PQ 5011 (Sanny, Brazil), making it possible to define some percentages between the different widths of the footprint, adapted from the study of [2,27]. Thus, the first footprint was defined as 1 × ADB, the second 1.3 × ADB, the third 1.5 × ADB (Figure 2).

### 2.7. Statistics

Descriptive statistics were obtained using the measures of central tendency, means (X) ± standard deviation (SD) and 95% confidence interval (95% CI). To verify the normality of the variables the Shapiro–Wilk test was used, considering the size of the sample. To evaluate the performance between the groups the ANOVA test (One Way) was used, with Bonferroni post hoc test. The confidence interval was calculated for the means found. The significance level adopted was *p* < 0.05. Cohen’s d was calculated as the difference between the mean divided by the pooled SD to estimate the effect size for between-lift comparison [28]. To check the effect size (Partial Eta squared: η2p), values of low effect (≤0.05), medium effect (0.05 to 0.25), high effect (0.25 to 0.50) and very high effect (>0.50) were adopted for ANOVA [28]. A d value < 0.2 was considered a trivial effect, 0.2 to 0.6 a small effect, 0.6 to 1.2 a moderate effect, 1.2 to 2.0 a large effect, 2.0 to 4.0 a very large effect, and ≥4.0 an extremely large effect [28]. Cohen’s “d” was calculated as the difference between the mean divided by the pooled SD to estimate the effect size for between-lift comparison [28]. The statistical treatment was done using the Statistical Package for the Social Science (SPSS), version 22.0. The level of significance was set at *p* < 0.05.

## 3. Results

Table 2 presents the results as mean and standard deviation with respect to the means of the velocity for the different grip amplitudes for loads of 30 and 50% of 1 RM. In Table 2, the results are presented.

A medium effect was observed, with significant difference being shown for BAD 1× in velocity average production with 30% (*p* = 0.03; η2p = 0.20) when compared to 1.3 × BAD. A medium effect was found for 30% (*p* = 0.77; η2p = 0.23) in velocity average propulsive when compared to BAD 1× to 1.5 × BAD. With respect to the other velocity variables for 30% and 50% of 1 RM over the different grip amplitudes, there were no significant differences. Nevertheless, there was a tendency for a high effect with respect to average velocity when compared to BAD 1× to 1.5 BAD to 50% (*p* = 0.18; η2p = 0.34) and BAD to 1.3 BAD and BAD 1× to 1.5 BAD to velocity peak respectively (*p* = 0.17; η2p = 0.42; *p* = 0.24; η2p = 0.34).

## 4. Discussion

The objective of our study was to evaluate the mean velocity averages for 30% and 50% of 1 RM with different handgrip amplitudes in PP athletes. Our hypothesis was that the 1.5 × BAD grip contributes to greater velocity, because it is the closest grip amplitude to that used by common athletes. The main result of this study showed that there was a significant difference in the VA of the barbell displacement at 30% of 1 RM with the 1 × BAD grip compared to 1.3 × BAD in PP athletes, but no significant difference compared to 1.5 × BAD. These findings did not answer our hypothesis; however, to our knowledge, this is the first study to relate different grip amplitudes and velocity averages in PP athletes. Since the bench press is an exercise widely used to assess velocity and force-power of the upper limbs [3,29], as well as being an exercise that is part of Powerlifting and PP competitions, we believe that our results are relevant to developing a better understanding of the variables involved in this sport.

On the other hand, when comparing the velocity variables in our study, it is worth noting that the VA for 30% of 1 RM was the only significant variable; according to García-Ramos et al. [30], VA can be considered a variable of self-power of linearity related to the load–velocity in supine exercise. Previous studies have presented the linear factor in the aggregation of the force–velocity relationship in the muscles involved in maximal multi-articular exercises [31], especially in straight supine exercises [26,32]. Therefore, would be interesting to understand the inter-relationships between velocity, force and load–velocity in PP.

There are still few studies that have evaluated the mechanical variables related to the performance of the adapted bench press performed by PP athletes. Loturco et al. [33] used the load–velocity relationship to predict the use of relative loads that can be part of the adapted supine, more specifically the monitoring of velocity averages to estimate training load. The authors reported that as loads were increased to >70% of 1 RM, the VA and VAP decreased, thus characterizing an effective contractile relationship between these variables in PP athletes.

In the present study, it was demonstrated that the mean velocity generated for 50% load of 1 RM was lower than that applied for 30% of 1 RM, regardless of the grip amplitude. According to Sanchez-Medina and Gonzalez Badillo [7], there is an inverse relationship between force and velocity. Even so, we identified some particularities in our study, where the MVP for the loads of 30% and 50% of 1 RM in relation to the different amplitudes of the grip was higher than that for velocities found in studies with regularly trained PP athletes, fighting athletes, and ruby players (MVP 30% 1 RM; 1.59 m/s and for 50% 1.06 m/s × 0.82 and 0.62 m/s PP and 0.80 m/s for 60% of 1 RM) [33,34]. Given this fact, we believe that the mechanisms for generating force and velocity seem to be the same, or that they represent a possible adaptation of the training to the neuromuscular system [35,36].

Other studies have used the velocity of displacement of the bar as a basis for predicting the relationship between the relative loads lifted in the squat, leg press, pully pull-up and the supine itself [3,37,38]. However, in the literature, the relationships between different grip amplitudes and the VA, VAP, and PV of bar displacement in the adapted supine is still unclear; to our knowledge, this is the first study that has performed such an evaluation. Thus, previous studies have investigated the performance of the bench press on the load and/or force–velocity relationship without worrying about specifying the ideal grip amplitude for the movement action, the main focus being on the reliability of using the averages of velocity on the basis of a linear and polynomial regression model to estimate the relative loads with respect to 1 RM in the bench press exercise [3,5,39].

It is worth noting that the VA and VAP generated with 1 × BAD for the 30% load of 1 RM were higher than the velocity achieved with the 1 × 3 BAD and 1 × 5 BAD grips. According to Gomo and Van den Tillaar [11] and Lockie et al. [12], narrower grip width is relevant for lifting lighter loads because it reduces the moment arm of the elbow and shoulder joints in the bench press. Additionally, according to Gomo and Van den Tillaar [11], using three different grip widths, the narrower grip tends to contribute to greater peak bar velocity. However, in contrast, in our study, the highest VP for the loads of 30% and 50% of 1 RM were found with a grip width of 1.3 × BAD even though it was not statistically significant.

## 5. Practical Applications

Due to the lack of studies that can guide researchers and sports scientists, as well as the practical performance of PP coaches, our results propose the use of the 1x grip of the BAD as an option for working the velocity of displacement of the bar when performed in the adapted bench press with loads at 30% of 1 RM in PP athletes. In addition, higher velocities of movement occurred with different grip widths. Therefore, grip width in training programs and exercises should be specifically chosen for their ability to influence sports performance.

## 6. Conclusions

In summary, our results indicated that the average bar movement velocity in Paralympic Powerlifting was higher with the 1 × BAD grip at 30% of 1 RM when compared to the 1.3 × BAD and 1.5 × BAD grips; similarly, there was no significant difference between 1 × BAD and 1.5 BAD, with the narrower grip width being suited to lifting lighter loads and having been shown to be accurate to the team’s technical staff and researchers involved in Paralympic powerlifting. On the other hand, as the load was increased to 50% of 1 RM the VA, VAP, and PV decreased when compared to 30% of 1 RM, to the point that the values of VA, VAP, and VP corresponding to the 1.3 × BAD and 1.5 × BAD grips were higher than 1 × BAD, except for the VA at 50% of 1 RM, in which the 1 × BAD grip was superior to the others. In fact, there was an inverse relationship between force and velocity and between the different amplitudes of the grip. Therefore, 1.5 × BAD was not superior to the other grip widths, and did not support our current hypothesis; in addition, the results revealed the importance of choosing the specific grip width for improving performance in Paralympic Powerlifting. Therefore, further research should use heavier loads in terms of the relation between velocity and the different amplitudes of the grip, for example 75–80%, or also examine 1 RM in Paralympic Powerlifting.

## Figures and Tables

**Figure 1 jfmk-06-00086-f001:**
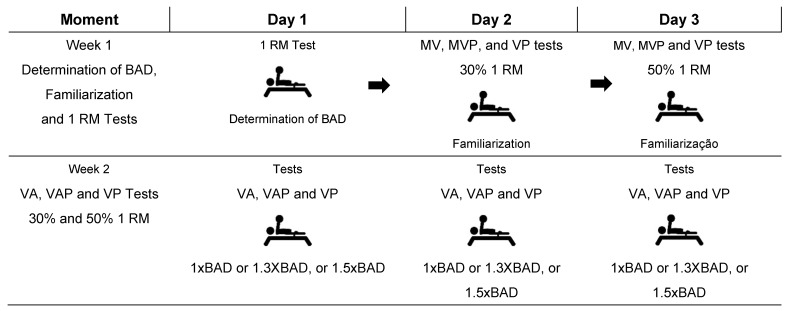
Experimental design. BAD: biacromial distance; VA: velocity average; VAP: velocity average propulsive; VP: velocity peak; 1 RM: 1-repetition maximum.

**Figure 2 jfmk-06-00086-f002:**
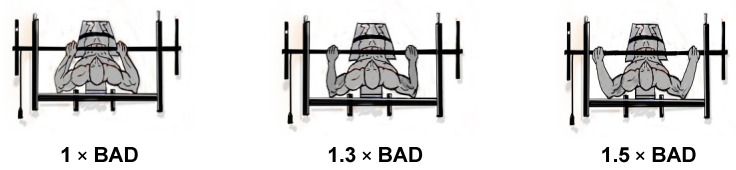
Demonstration of the various footprint widths.

**Table 1 jfmk-06-00086-t001:** Characterization of subjects.

	(Mean ± SD)
Age (years)	25.40 ± 3.30
Body Mass (Kg)	70.30 ± 12.15
Experience (years)	2.45 ± 0.21
Grip width 1 × Biacromial (cm)	42.83 ± 12.84
Pick up width 1.3 × Biacromial (cm)	55.68 ± 16.70
Catch width 1.5 × Biacromial (cm)	63.20 ± 18.96
1 RM bench press test (Kg)	117.40 ± 23.37 *
1 RM/body weight	1.67 ± 0.28 **

* All athletes with loads that keep them among the top 10 in their categories at the national level. ** Values above 1.4 in the Bench Press would be considered elite athletes, according to Ball and Wedman [19].

**Table 2 jfmk-06-00086-t002:** Velocity average (VA); velocity average propulsive (VAP); and velocity peak (VP) (mean standard deviation) in relation to different footprint widths for 30% and 50% 1 RM.

		BAD 1× (A) X ± SD (CI 95%)	BAD 1.3× (B) X ± SD (CI 95%)	BAD 1.5× (C) X ± SD (CI 95%)	A vs. B	A vs. C	B vs. C	*p*	η2p
30% RM	VA (m/s) VAP (m/s) VP (m/s)	1.16 ± 0.14 (1.07–1.26) 1.59 ± 0.31 (1.44–1.74) 1.63 ± 0.22 (1.49–1.72)	1.00 ± 0.17 (0.90–1.09) 1.55 ± 0.31 (1.40–1.70) 1.73 ± 0.24 (1.59–1.86)	1.06 ± 0.09 (0.97–1.16) 1.52 ± 0.16 (1.37–1.67) 1.70 ± 0.08 (1.57–1.84)	*p* = 0.04 * d = 0.20 *p* = 0.92 d = 0.17 *p* = 0.55 d = 0.41	*p* = 0.31 d = 0.84 *p* = 0.77 d = 0.48 *p* = 0.72 d = 0.43	*p* = 0.58 d = 0.48 *p* = 0.95 d = 0.12 *p* = 0.96 d = 0.14	0.04 0.77 0.77	0.20 # 0.23 # 0.23
50% RM	VA (m/s) VAP (m/s) VP (m/s)	0.84 ± 0.10 (0.79–0.88) 1.00 ± 0.09 (0.92–1.08) 1.19 ± 0.07 (1.14–1.24)	0.81 ± 0.02 (0.76–0.85) 1.06 ± 0.11 (0.98–1.14) 1.26 ± 0.06 (1.20–1.31)	0.78 ± 0.05 (0.73–0.82) 1.00 ± 0.13 (0.92–1.08) 1.25 ± 0.08 (1.20–1.30)	*p* = 0.65 d = 0.39 *p* = 0.55 d = 0.54 *p* = 0.17 d = 0.95	*p* = 0.18 d = 0.73 *p* = 0.99 d = 0.04 *p* = 0.25 d = 0.073	*p* = 0.63 d = 0.71 *p* = 0.50 d = 0.49 *p* = 0.98 d = 0.10	0.18 0.17 0.24	0.34 ## 0.42 ## 0.34 ##

* *p* < 0.05 ANOVA (one-way), and Bonferroni post hoc. ES: effect size, # medium effect, ## high effect; BAD: Biacromial distance; VA: velocity average; VAP: velocity average propulsive; and VP: velocity peak. Thirty percent of one repetition maximum 30% 1 RM; fifty percent of 1 RM; CI: confidence interval.

## Data Availability

The data that support this study can be obtained from the address: www.ufs.br/Department of Physical Education (accessed on 12 October 2021).

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
