# Peer review of "Analysis of Grip Amplitude on Velocity in Paralympic Powerlifting"

_jfmk, 2021, doi:10.3390/jfmk6040086_

Round 1

Reviewer 1 Report

the work presented shows an element of novelty, it is well constructed, a revision of the English form would be advisable.

One point to review, why was the relatively low percentages of 1RM chosen? And even following this scheme, another test could be foreseen, for example at 70-75% and/or a check also at 1RM, perhaps simulating a competition situation.

Author Response

Dear Reviewer, 

Reviewer 1

The work presented shows an element of novelty, it is well constructed, a revision of the English form would be advisable.

English was revised as requested.

One point to review, why was the relatively low percentages of 1RM chosen?

At first we thought of evaluating loads that could be displaced close to 1 m/s. An alternative to heavy loads, however, we intend to use heavy loads in a future study.

And even following this scheme, another test could be foreseen, for example at 70-75% and/or a check also at 1RM, perhaps simulating a competition situation.

As this is a study more focused on training, we followed the speed as mentioned in the previous item, and thus the percentages could not be higher

Kind regards, 

Reviewer 2 Report

Some of the Key words are not found as descriptors in health sciences.

Why was the 30 and 50% load chosen for the objective? Why were other different loads not measured?

The material and methods section should begin with the study design. In fact, it starts talking about the second week without mentioning anything about the first week (which is then reflected in the procedure section).

Inclusion and exclusion criteria are not shown.

In Table 2, I suggest including the exact p value and effect size for pairwise comparisons.

Paragraph 207-211 repeats the same information as Table 2.

It would also be interesting to include the distance the bar moves in each grip.

In the conclusion of the paper it is stated that the velocity is greater for 1xBAD at 30% compared to the other grips. However, the statistics show no difference in speed with the 1.5xBAD.

The conclusion seems to be a repetition of the result set. The conclusion of this study should be stated here.

This would be practical application, not conclusions: "Due to the lack of studies that can guide researchers and sport scientists, as well as the practical performance of PP coaches, our results propose the use of the 1x grip of the BAD as an option to work the velocity of displacement of the bar when performed in the adapted bench press with loads at 30% of 1 RM in PP athletes".

This would be justification and should be stated in the introduction: "Understanding that the VA is characterized since the beginning of the concentric action until the final moment of arms extension and being this a common action to be evaluated in competition, the VA tends to be an important variable to measure the velocity of displacement of the bar in the adapted supine with loads from 30% to 40% of 1 RM in athletes of the PP."

Author Response

Dear Reviewer, 

Reviewer 2

Some of the Key words are not found as descriptors in health sciences.

Adjusted Line 49 Paralympic Powerlifting, grip Width, Relative Load and Velocity.

Why was the 30 and 50% load chosen for the objective? Why were other different loads not measured?

At first we thought of evaluating loads that could be displaced close to 1 m/s. An alternative to heavy loads, however, we intend to use heavy loads in a future study.

The material and methods section should begin with the study design. In fact, it starts talking about the second week without mentioning anything about the first week (which is then reflected in the procedure section

Adjusted  Line 97-98

In the first week the subjects were submitted to a familiarization session, 1 RM test and definition of the biacromial distance as the base for the different grasp amplitudes.

Inclusion and exclusion criteria are not shown.

Adjusted Line 110-112

As an inclusion criterion, with at least one year of competitive experience in the sport and the athlete had to have participated in at least one competition at the national level in the period of 1 year.

In Table 2, I suggest including the exact p value and effect size for pairwise comparisons.

We include the exact "p" value as suggested.

Paragraph 207-211 repeats the same information as Table 2.

Adjusted Line 209-216

Medium effect with significant difference were found for BAD 1X in average velocity production with 30% (p= 0.03; hp2= 0.20) when compared to 1.3X BAD. Medium effect were found for 30% (p= 0.77; hp2 =0.23) in velocity average propulsive when compared to BAD 1X to 1.5X BAD. Regarding the other velocitys for 30% and 50% of 1 RM over the different grip amplitudes there was no significant difference. Nevertheless, there was a tendency for high effect in average velocity when compared to BAD 1X to 1.5 BAD to 50% (p= 0.18; hp2 =0.34) and BAD 1X to 1.3 BAD and BAD 1X to 1.5 BAD to velocity peak respectively (p= 0.17; hp2 =0.42; p= 0.24; hp2 =0.34).  

It would also be interesting to include the distance the bar moves in each grip.

We understand and appreciate your feedback. However, we don't have the distance to add. It's a fact we'll explore in the future!

In the conclusion of the paper it is stated that the velocity is greater for 1xBAD at 30% compared to the other grips. However, the statistics show no difference in speed with the 1.5xBAD.

Adjusted Line 283-288

Our results indicated that, the average bar movement velocity in Paralympic Powerlifting was higher with the 1x BAD grip at 30% of 1 RM when compared to the 1.3x BAD and 1.5x BAD grips, same there having no significant difference in between 1x BAD and 1.5 BAD, indicated that narrower grip width, is related to lifting lighter loads and shown to be accurate to team’s technical staff and researchers involved in Paralympic powerlifting.

The conclusion seems to be a repetition of the result set. The conclusion of this study should be stated here.

Adjusted Line 283-298

In summary, our results indicated that the average bar movement velocity in Paralympic Powerlifting was higher with the 1x BAD grip at 30% of 1 RM when compared to the 1.3x BAD and 1.5x BAD grips, same there having no significant difference in between 1x BAD and 1.5 BAD, the narrower grip width, is related to lifting lighter loads and shown to be accurate to team’s technical staff and researchers involved in Paralympic powerlifting. On the other hand, as the load was increased to 50% of 1 RM there was a tendency the VA, VAP, and PV decreased when compared to 30% of 1 RM, to the point that the VA, VAP, and VP generated by the 1.3x BAD and 1.5x BAD grips were higher than 1x BAD, except for the VA at 50% of 1 RM in which the 1x BAD grip was superior to the others. Fact that an inverse relationship between force and velocity and between an relation to the different amplitudes of the grip. Therefore, 1.5x BAD no was superior to the others grip width and did not answer our hypothesis current, in addition, the results revealed the importance of choosing the specific grip width for improvement of performance in Paralympic powerlifting. Therefore, further research must use heavier loads between an relation the velocity to the different amplitudes of the grip, for example 75-80% or check also at 1RM in Paralympic powerlifting

This would be practical application, not conclusions:

Adjusted Line 274-280

Due to the lack of studies that can guide researchers and sport scientists, as well as the practical performance of PP coaches, our results propose the use of the 1x grip of the BAD as an option to work the velocity of displacement of the bar when performed in the adapted bench press with loads at 30% of 1 RM in PP athletes. In addition, higher velocities of movement occurred during different grip widths, so grip witdh in training programs and exercises should be specifically chosen for can influence sports performance.

This would be justification and should be stated in the introduction:

Adjusted Line 74-78 "Understanding that the VA is characterized since the beginning of the concentric action until the final moment of arms extension and being this a common action to be evaluated in competition, the VA tends to be an important variable to measure the velocity of displacement of the bar in the adapted supine with loads from 30% to 40% of 1 RM in athletes of the PP."

Kind regards, 

Round 2

Reviewer 2 Report

I recommend the authors to continue working along these lines, using higher loads.